# Megastudies: A New Approach to Reducing Vaccine Hesitation Worldwide

**DOI:** 10.3390/vaccines11010133

**Published:** 2023-01-06

**Authors:** Lian Yu, Jiaqi Qiao, Wai-Kit Ming, Yibo Wu

**Affiliations:** 1Health Care System Reform and Development Institute, School of Public Health, Xi’an Jiaotong University, Xi’an 710061, China; 2Jinhe Center for Economic Research, Xi’an Jiaotong University, Xi’an 710061, China; 3Department of Infectious Diseases and Public Health, Jockey Club College of Veterinary Medicine and Life Sciences, City University of Hong Kong, Hong Kong 999077, China; 4School of Public Health, Peking University, Beijing 100191, China

**Keywords:** vaccine hesitation, megastudies, RCTs, social factors, quality control, justice

## Abstract

Vaccine hesitancy is a considerable obstacle to achieving vaccine protection worldwide. There needs to be more evidence-based research for interventions for vaccine hesitancy. Existing effectiveness evaluations are limited to one particular hypothesis, and no studies have compared the effectiveness of different interventions. A megastudy takes a large-scale, multi-intervention, uniform participant and the same evaluation criteria approach to evaluate many interventions simultaneously and find the most effective ones. Therefore, megastudies can help us find the most effective interventions for vaccine hesitancy. Additionally, considering the complex causes of vaccine hesitancy, we design interventions that involve social factors in megastudies. Lastly, quality control and justice are critical issues for megastudies in the future.

## 1. Introduction

The World Health Organization defines *vaccine hesitancy* as a “delay in acceptance or refusal of safe vaccines despite the availability of vaccine” [1]. Infectious diseases are an ever-present threat to humans, while vaccination prevents over 20 life-threatening diseases, helping people of all ages live longer, healthier lives. Immunization currently prevents 3.5–5 million deaths yearly from diseases such as diphtheria, tetanus, pertussis, influenza, and measles [2]. According to research on the relationship between vaccine refusal and vaccine-preventable disease outbreaks in the United States, 70.6% of measles-infected individuals were unvaccinated in measles outbreaks from 2000 to 2015 [3]. Additionally, vaccination works as individual immunity and achieves the protective effect with a significantly vaccinated population; the higher the coverage and the faster coverage is achieved, the better the vaccine’s protection [4,5].

The COVID-19 (coronavirus disease 2019) global pandemic is accompanied by ultra-contagiousness, a rapid mutation rate, and a lack of specific drugs, making vaccines a key measure for reducing its damage. According to a target population size for COVID-19 vaccination, the expected global vaccination population for COVID-19 is estimated to be 5.8 billion, representing 74.4% of the worldwide population [6]. However, vaccine hesitation is a severe obstacle to this goal [7,8]. According to an Oxford University research institute, by 27 October 2022, only 68.4% of the world population had received at least one dose of a COVID-19 vaccine [9]. In January 2019, the WHO (World Health Organization) declared vaccine hesitancy among the top 10 threats to global health [1].

Vaccine hesitancy occurs in response to all sorts of vaccines and exists to varying degrees around the world [10,11,12]. SAGE (World Health Organization Strategic Advisory Group of Experts on Immunization) defined vaccine hesitancy as “complex and context-specific,” “varying over time and place,” and “vaccine specific” [13]. Worse, fragile groups, such as people with chronic diseases, the elderly, and pregnant women, usually hesitate to take vaccines [14,15,16]. In addition, marginal groups in deep need of vaccine protection (e.g., minority groups, migrants, or refugees) may be more vaccine-hesitant, which is worrying [17,18,19]. Multiple researchers have discussed vaccine hesitancy’s causes and influencing factors [20]. They revealed that causes of vaccine hesitancy, including worrying about vaccines’ safety [21]; inconvenience in accessing vaccines [1]; mistrust of the medical system, government, or a new vaccine [22]; religious belief [23]; personality [24]; complacency (underestimated risk of infection) [25]; conspiracy thinking [26]; negative information about vaccines spread on social media [27]; and so on.

Interventions to reduce vaccine hesitancy are urgently needed. Studies on the causes and influencing factors can guide us to strategies and skills to reduce vaccine hesitancy. Many studies have brought up suggestions based on vaccine hesitancy cause analysis [28]. For example, a meta-analysis on measles and HPV (human papillomavirus) vaccines introduced nine vaccine promotion intervention domains. They include education (providing education on vaccination, disease, and how vaccines work); on-site vaccination (providing vaccines at the workplace or places of worship); financial incentives; free vaccination; institutional recommendation (a recommendation made by the institution that person works at especially for healthcare providers); healthcare provider recommendation; a reminder and recall; gain vs. loss framing of the vaccine; and vaccine champions (an institutionally appointed champion to encourage vaccination) [29].

In the research on different vaccines, researchers believe we need strategies on the individual, provider, health system, and national levels [5,30]. Some researchers find specific theoretical models (interventions based on the health belief model, the theory of planned behavior, and the 3C model (complacency, convenience, and confidence model)) could provide us with excellent guidance to reduce vaccine hesitancy in pregnant women [31]. The idea of designing community engagement strategies aiming to build public trust was brought up [22]. In addition, many researchers have evaluated the effects of interventions in rigorously controlled studies, and many interventions proved effective, such as individually education, sending reminder messages, and financial incentives [28,32,33].

We already know so many inventions for vaccine hesitancy, but which intervention or mix-program is most effective in promoting vaccination? The cost of vaccine hesitancy to global health and economies is so great that we need more than a random practice of interventions or the selection of interventions based only on the evidence of scattered studies. Instead, we need to find the most effective interventions for vaccine hesitancy based on more rigorous evidence.

Nonetheless, RCTs (randomized controlled trials) on vaccine hesitancy have only examined the effects of one intervention limited to a single location and a tiny population and are highly specific. Given the different research designs and participants, we cannot compare the effectiveness of inventions in different studies. Additionally, obtaining information particular interventions one by one would take considerable time.

Therefore, we have an urgent task: compare various interventions’ efficacies to find the most effective ones.

## 2. Emerging Megastudies in Behavioral Study

In response to these problems, a megastudy offers a new and attractive option. Megastudies have been applied in psychology, linguistics, and computer science for years [34,35,36,37]. At its core, it is about different researchers using the same dataset and searching for the optimal outcome to solve a common problem. It is currently applied in behavioral science.

Insufficient physical exercise is a substantial public health issue. In this field, researchers test different intervention ideas in different samples using different methods over different time intervals. The problem is that comparing those conclusions from individual investigations is difficult. A study published in *Nature* in 2021 described megastudies on exercise behavior. Researchers realized that the “one-apple-at-time” approach is inefficient in advancing behavioral science. “Typically, individual research studies are designed to establish the validity of a single idea rather than to assess its efficacy relative to other theoretically informed approaches in a particular policy context. We propose that the megastudy approach surmounts this and many other obstacles to developing optimal behaviorally informed policy interventions” [38].

With 61,293 participants, small independent teams of 30 scientists from 15 different US universities designed and implemented 54 interventions (designed by different independent teams) to encourage exercise for four weeks and then identified the best intervention with the same standard. The interventions practiced in this megastudy include a bonus for returning after missed workouts, exercising social norms shared (high and increasing), free audiobook provided, planning fallacy described and planning revision encouraged, rigidity reward, and others in a total of 54 interventions.

This megastudy on exercise included a placebo control group in which participants received a little reward to enter an experimental condition and no other incentives. Researchers set the planning, reminders, and micro-incentives as the baseline, then tested the other 53 experimental conditions. “Where a typical RCT study would develop an intervention based on one hypothesis, megastudies trying many interventions at once—as was done in the megastudy—could speed up scientific discovery” [39]. However, the most significant increase (27% increase in exercise and 16% increase compared to the baseline) occurred in the condition of “a bonus for returning to the gym after a missed workout,” which differs greatly from the prediction [38].

Is this research approach suitable for the pressing question of vaccine hesitancy? After the exercise behavior megastudy, Milkman conducted a study on vaccine hesitancy in 2020 [40]. Studies have shown that only 79% of people who intended to receive the influenza vaccine did so [41]. Follow-up failure results from forgetfulness, failure to expect and plan barriers, and lack of motivation [42]. Psychological reminders have the potential to bridge these “intention-action gaps” [43]. Hence, researchers randomly assigned 689,693 Walmart pharmacy patients to receive one of 22 text reminders using various behavioral science principles to nudge flu vaccination or to a business-as-usual control condition where patients received no messages. The most effective messages reminded patients that a flu shot was waiting for them, and they delivered reminders on multiple days. Like the megastudy on exercise, neither experts nor lay people expected the best-performing treatment, underscoring the value of simultaneously testing many nudges in a highly powered megastudy [40].

In October 2022, Duckworth and Milkman summarized the megastudy approach in *A Guide to Megastudies* [44]. They described its origins, implementation steps, strengths, conditions, limitations, and future. These presentations allow us to fully understand this approach and provide a desirable option for reducing vaccine hesitancy. Duckworth and Milkman define megastudies in behavior studies as “a massive field experiment in which many different treatments are tested synchronously in one large sample using a common, objectively measured outcome” [37,44]. It typically takes the form of independent research teams developing sets of treatment(s) and control conditions (“sub-studies”), with participants randomly assigned across all of them, such as what Duckworth and Milkman illustrate in the following Figure 1 [44].

Duckworth and Milkman briefly discussed the advantages and disadvantages of megastudies. Next, we must discuss the advantages and disadvantages of megastudies and the challenges and limitations of applying megastudies to address vaccine hesitancy.

## 3. Strengths and Weakness

Faced with vaccine hesitancy, a significant global public health problem, we need to adopt random interventions that have only been proven “effective”. Traditional single RCT studies can provide more valid evidence than other studies (e.g., cross-sectional studies), but their results only validate one particular hypothesis in the research. Even though researchers have carried out corresponding studies around the same topic, the results are difficult to compare because of their different study designs and participants. In contrast, in megastudies, participants for each intervention come from a shared pool. Each intervention takes the same period and is assessed with the same set of criteria, which allows us to draw conclusions about which behavioral interventions are most reliable.

The strengths of megastudies over traditional individual RCTs include the following:**Optimal Choice:** compared to traditional single RCTs, megastudies can find the most effective intervention rather than one effective intervention with direct evidence because the researcher applies many interventions within the same population, time, and condition.**Less Time:** Compared to many single RCTs applied by different teams in different periods, megastudies can find the most effective interventions to increase vaccination coverage relatively quickly. This is vital when a pandemic breaks out.**Broad Vision:** Compared to traditional single RCTs, megastudies may find unexpected outstanding interventions. Because traditional RCT studies usually test hypotheses, researchers suppose the most likely option and do not try those in which the researcher has little confidence. Hence, they may miss interventions that appear less promising but are, in fact, highly effective [38,40]. In megastudies, different teams present different interventions and may apply interventions of which they are less confident [44].**Complete Results:** Megastudies compare the efficacy of different interventions and find the optimal approach so researchers do not have to worry about obtaining invalid results for specific measures and publishing invalid results, which is also essential for our guidance. In independent studies, researchers may not publish invalid results that cause misinformation [44].**Inclusion:** Megastudies can involve more researchers. Different teams from different disciplines and schools of thought can make the study more diverse [44]. However, if the selection criteria and procedures are inappropriate, it may also cause inclusion problems. Who will be invited, and who will not? Therefore, we also need to design open and scientific procedures to recruit researchers.

Nonetheless, compared with traditional RCTs, some disadvantages appear in megastudies.

**Cost:** The most apparent burden of megastudies may be the considerable cost. For example, the exercise study by Milkman et al. costs $2.6 million to implement the intervention. However, given the enormous health and economic losses to humans caused by vaccine hesitancy, megastudies in the vaccine hesitancy field are urgent. Secondly, megastudies are costly compared to one traditional RCT but may save much money compared with numerous independent studies. Duckworth and Milkman mentioned that high costs might make it challenging to conduct megastudies and difficult to replicate [44].**Applicability:** Conclusions from megastudies cannot be extrapolated. The evidence level from megastudies is not higher than RCTs: we should know that a megastudy remains an RCT. Megastudies differ from a traditional single RCT in evidence power and research purposes. Megastudies help us compare the effectiveness of many interventions, finding the best one from many interventions (there are both baseline and placebo groups inside) rather than finding the effective one in two options (the placebo group).

Despite these disadvantages, megastudies are still a desirable option in the face of the urgent need to reduce vaccine hesitation. However, there are some critical issues if we want to reduce vaccine hesitation in megastudies.

## 4. Challenges and Solutions

Vaccine hesitancy is not only a behavioral problem and complex social phenomenon [45]. The interventions in physical exercise and influenza megastudies focus on the gap between intention and action [29]. Interventions in two megastudies for behavior change were mainly “Nudges”: planning, bonus, exercise commitment contracts, sharing social values or norms in the community, fitness questionnaires, and different urging messages [38,40]. However, many people still have not made a firm commitment to vaccination. Can megastudies be conducted on these populations? Let us examine different causes that influence vaccination intentions.

**Safety and Effectiveness:** Worrying about the safety and effectiveness of vaccines is a common factor of vaccine hesitancy [20.21]. Many people worry about the damage caused by vaccines, especially when adverse events are reported [46,47]. Among those concerned about vaccine safety, some are concerned about additional risks because of their clinical conditions, such as pregnant women [16,30], the elderly [16,48], and patients with chronic diseases [49,50]. In recent years, vaccine hesitancy has closely been related to widespread rumors about infertility or autism, which proved totally false [29]. In addition, suspicion of vaccine effectiveness matters in vaccine hesitation, especially facing a new vaccine, the vaccine does not work as expected or few studies have been conducted to demonstrate its efficacy and safety in specific populations at risk [46,51,52,53].**Social Factors:** Except for the clinical factors, many social factors are tangled with the worry about vaccines. Sometimes, vaccine hesitancy is provoked by racism, inequality, social exclusion, and religious ideas [17,18,19,23]. Education, power structures and gender inequalities in the family may also trigger vaccine hesitancy. For example, the survey revealed that fathers with limited education might have a more significant influence on preventing their children from being vaccinated than mothers with more education. The blocking effect of maternal grandfathers was more pronounced [54]. Additionally, social media drives the widespread dissemination of rumors about vaccines [27,55,56,57]. Finally, some suspect vaccines are a conspiracy of other countries, governments, or big companies [26,54]. In some cases, resistance reflected not specific concerns about the vaccine but rather a convergence of broader social factors [45].**Diversity:** The reasons for vaccine hesitancy are diverse. Vaccine hesitancy is related to people’s characteristics, such as race, health conditions, age, religion, education, political and economic status, region, and so on [20,51]. Studies have shown that differences in vaccination rates between China and the United States is slight. However, the reasons for vaccination and concerns about vaccines are very different. Therefore, it is necessary to develop different comprehensive intervention strategies for different countries [58]. Therefore, causes for vaccine hesitancy may differ for populations, and the effective interventions for them may differ.

These problems cannot be solved only by nudges. If there are questions about the safety, efficacy, and accessibility of the vaccine, we should solve them in the development, production, and distribution process [59,60]. When discussing vaccine hesitancy, it is supposed to be safe and accessible [1]. However, even if vaccines are safe and effective, people may not believe it; even if vaccines are accessible, people may not know. We can intervene in these issues.

We already have many recommendations to address vaccine hesitation provoked by various causes. Some suggestions are about policies and institutional change, such as introducing more transparency into policy decision-making before immunization programs, more public participation in vaccine programs, providing up-to-date information to the public and health providers about the rigorous procedures undertaken before the introduction of new vaccines, and through diversified post-marketing surveillance of vaccine-related events [61]. Although many studies have shown that negative messages on social media increase vaccine hesitation, some studies show that open media use can increase new crown vaccinations. Therefore, we can make individualized and targeted strategies for COVID-19 vaccination and disseminate the vaccination information to different media use groups [62]. Nevertheless, many suggestions are easy to apply and proved effective, such as individual education, a loss–benefit framework, a recommendation from health service providers, and so on [28,29]. Then, we need to know which is most effective for a particular group or cause. In addition, we can design new intervention programs according to existing research and use megastudies to find the most effective one.

For example, scholars have suggested several plans for the conflict between fasting and vaccination for Muslims: staggering the time of vaccination with the time of fasting (e.g., providing vaccine at night, during special Ramadan nightly prayers, Taraweeh), having religious leaders explain that there is no contradiction between fasting and vaccination, and setting up vaccination sites outside mosques [23]. How effective are these methods? Which one, or what combination, will be most effective? We can only know with megastudies. The answer may differ from what we might expect [63].

There are many long-standing, systemic social problems behind vaccine hesitation, which are difficult to change in the short term. However, we can still promote vaccination before completely solving the related social problems. For example, immigrants may distrust local governments because of social exclusion. Hence, they may understand vaccines as a conspiracy. We cannot change discrimination against immigrants in the short term, nor can we change immigrants’ attitudes toward the government in the short term. However, we may change immigrants’ perceptions of vaccines via transparent information or change their perceptions of vaccines via key opinion leaders in the migrant community. In addition, there are many ways to address rumors and conspiracy theories on social media that may reverse false beliefs, such as social media postings to dispel rumors, celebrities promoting the safety of vaccines, and medical professionals explaining the safety of vaccines.

The core strength of megastudies is that they simultaneously compare multiple interventions in the same population. We can apply it with different interventions. Duckworth and Milkman have talked about the future of megastudies. They believe that megastudies could test more social and experiential interventions and aim to treat participants for extended time horizons [43]. However, they did not talk in detail. In the future, to reduce vaccine hesitancy, the involvement of social science scholars may be essential in megastudies. There is no conflict between the social sciences and megastudies: sociological and political science researchers have used megastudies to reduce people’s partisan hatred [64].

In response to the heterogeneity behind vaccine hesitancy, we propose to find optimal interventions for specific populations and causes rather than searching for a universal plan. We could, for example, conduct more thorough studies targeting groups such as the elderly, the chronically ill, immigrants, and minorities. In addition, we can conduct megastudies that target a particular cause of vaccine hesitation, such as addressing concerns about safety or social media rumors, to find the optimal intervention for a particular worry.

Still, we must admit that not all strategies that may reduce vaccine hesitancy can be tested in megastudies. Duckworth and Milkman admit that policy solutions are amenable to examination by field experiment or megastudy, and, of course, many of the most efficacious solutions may not be [44]. For example, establishing a transparent vaccine research mechanism and a rigorous safety system is crucial to reducing vaccine hesitation, but we may not verify its effectiveness in megastudies.

## 5. Further Challenges for Megastudies in Future

We should involve more social factors in megastudies on vaccine hesitancy. Meanwhile, it is necessary to take megastudies in different populations and social contexts or focus on vaccine hesitancy provoked by different causes. However, new challenges may then appear if we do so. For example, interventions in studies may be more complex, labor-intensive, and longer processes. Moreover, it may be challenging to access marginal groups, such as the elderly who do not use a smartphone, migrants who do not trust the medical system, and women who cannot read. Therefore, we must discuss the following significant issues:

**Quality Control:** Megastudies contain complex procedures and massive data. How can we manage it? Would falsification happen?

No study can eliminate errors, nor can we eliminate the possibility of falsification. However, by analyzing the study design, people may have a weaker incentive to falsify megastudies. In a single RCT study, people want to verify that an intervention is effective. If the results are negative, it may mean that the study has limited value or even no possibility of publication. Thus, researchers may be more motivated to falsify data and get valid results. In megastudies, as mentioned earlier, invalid results are not terrible. Therefore, even if a team designs an intervention that is ineffective or not the most effective, their work is a significant part of the overall study.

However, we recognize participants may cut corners. Sometimes, it could result from irregular training, and sometimes it may result from an overloading job. Therefore, it would be better to have stable sites or platforms. For example, gyms and pharmacies are excellent places, and hospitals, community health centers, or schools may be suitable. In addition, applying interventions via digital platforms may save labor and prevent falsification.

**Ethical Issues:** In many cases, vaccine hesitancy is related to a vulnerable position. Therefore, justice is a crucial ethical issue in the hesitation toward vaccines, and there would be a necessary concern about justice in megastudies on vaccine hesitancy.

Achieving adequate vaccine protection requires the participation of everyone. However, some groups may lack understanding about vaccines because of their marginal status (e.g., the elderly or disabled, or illegal immigrants). Similarly, because of their marginal status, it may be challenging for socioeconomically disadvantaged populations to take part in megastudies.

Even if one person from a marginal group participates in the study, it could be challenging for him or her to complete the interventions, some of which may require participants to have specific competencies (e.g., literacy, smartphone use, access to the Internet). In addition, factors such as time, financial, technical, or cognitive thresholds may exclude some vulnerable populations. We may not involve these participants in the study and thus cannot find the best way to intervene with vaccine hesitancy for marginalized populations or come up with the wrong way to intervene. Injustice could cause unaddressed vaccine hesitancy in marginalized populations and create new health inequities [65]. Without consideration of these factors, the strengths of megastudies may decrease, even causing health inequalities in the long term. Therefore, some details of the study design—such as simplifying the interventions, flexible time, and accessibility—could be crucial for a megastudy.

## 6. Conclusions

Vaccines have saved millions of lives against infectious diseases, and vaccine hesitancy is one of the most significant impediments to the protective effect of vaccines. By establishing which interventions are the most effective in reducing vaccine hesitancy, megastudies provide an irreplaceable approach to addressing this urgent problem. In the future, we must continue to improve the design of megastudies, expand the areas of application of megastudies, improve the procedures for administering megastudies, and develop ethical norms for megastudies.

Nevertheless, megastudies may not eliminate vaccine hesitancy, much less eliminate all the problems that prevent vaccination. Sometimes, the safety, efficacy, and accessibility of vaccines are not resolved [51,59], and sometimes we need laws and policies or systematical change [5,30,66]. Last but not least, it is necessary to reflect on the relationship between science and society [46]. However, even if vaccine hesitancy cannot be eliminated, megastudies may still select the most effective interventions to reduce vaccine hesitancy as much as possible. Although not the complete picture, a megastudy will be a crucial and irreplaceable piece in reducing vaccine hesitancy.

## Figures and Tables

**Figure 1 vaccines-11-00133-f001:**
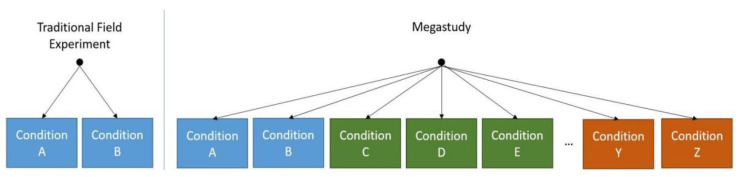
By Duckworth and Milkman, in *A Guide to Megastudies*. Left panel: traditional field experiments randomly assign participants to multiple conditions (e.g., conditions A and B), testing a limited number of related hypotheses. Right panel: megastudies randomly assign participants to a larger set of treatments often clustered by sub-study (different colors indicate different sub-studies), each testing potentially unrelated hypotheses (e.g., conditions A, B, C, D, etc.) [44].

## Data Availability

Not applicable.

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
