# Peer review of "Megastudies: A New Approach to Reducing Vaccine Hesitation Worldwide"

_vaccines, 2023, doi:10.3390/vaccines11010133_

Round 1

Reviewer 1 Report

I read carefully the paper titled " Megastudies: A New Approach to Reduce Vaccine Hesitation Worldwide " which argues for the need for megastudies on vaccination hesitancy.

While the topic is interesting, several corrections are needed.

Other reasons for hesitation should also be explained with appropriate reasons in the introduction. Among these agophobia, miscommunication by governments and attempts to minimize side effects rather than describe them with cold objectivity have played a decisive role. This should emerge in the paper. [cfr Nioi, Matteo, and Pietro Emanuele Napoli. "The Waiver of Patent Protections for COVID-19 Vaccines during the ongoing Pandemic and the Conspiracy Theories: Lights and Shadows of an Issue on the Ground." Frontiers in Medicine 8 (2021); Nioi, Matteo, et al. "Dual corneal-graft rejection after mrna vaccine (Bnt162b2) for covid-19 during the first six months of follow-up: Case report, state of the art and ethical oncerns." Vaccines 9.11 (2021): 1274..].

Another problem that has emerged in recent years is the falsification of data in scientific studies and the mass disclosure of the same data. Are the authors sure that the data from megastudies can then be useful in convincing people to get vaccinated?

Although the pro-vaccine approach has represented a scientific dogma in the last two years, data that has recently emerged have highlighted the presence of various controversial issues, in particular on efficacy and pre-market tests, and there has been a radicalization on the issue (politics, not scientific). The authors are advised to generalize when discussing vaccination hesitancy and to remain generic also in the conclusions reached. (discussing vaccines and not specifically COVID-19 vaccines). This is also because it seems that in various countries a universal vaccination policy will no longer be supported for the prevention of the SARS-19 disease.

Given the premises, it should be better explained how, according to the authors, mega-studies can influence those reluctant to get vaccinated given the points noted above.

Finally, references should be implemented.

Best regards.

Reviewer 2 Report

N/A

Author Response

We are grateful for your time and effort! 

Reviewer 3 Report

The comment “Megastudies: A New Approach to Reduce Vaccine Hesitation Worldwide” by Lian Yu et al, highlights the role of vaccine hesitancy as an obstacle to achieving optimal levels of vaccination coverage worldwide for infectious disease control.

The authors propose the use of megastudies to establish the most effective interventions to reduce vaccination hesitancy and provide a solution to this important issue.

The topic appears to be very relevant, but the manuscript requires several corrections and insights.

In the Introduction section, the authors refer “to all sorts of vaccines”, but only the measles outbreak and COVID-19 pandemic are mentioned.

Reference 2 refers to measles and pertussis. The authors could also refer to pertussis vaccination in the text and mention other vaccine-preventable diseases (HPV, hepatitis B, rubella, …..).

Some statements in the text are not appropriately supported by the cited literature.

For example, citation 3 is not appropriate.

The sentence in line 42 "The problem of vaccine hesitation is a severe obstacle to this goal" refers to lines 37-40. 

Please, revise the sentence of lines 40 to 42 according to the reported reference.

In line 45 the authors refer to complex reasons for vaccine hesitation, but these reasons are not mentioned, except concerning minority groups and migrants. “Complacency, inconvenience in accessing vaccines, and lack of confidence are the key reasons underlying hesitance according to the WHO (Reference 1).

Citations are missing (page 2, lines 47-49) 

In line 54, the authors declare that there is a severe lack of empirical studies on interventions to reduce vaccine hesitance. The authors could for example consider some interventions included in the Catalog of interventions addressing vaccine hesitancy provided by the European Center for Disease Prevention and Control (https://www.ecdc.europa.eu/sites/default/files/documents/Catalogue- interventions-vaccine-hesitancy.pdf ) and modify the statement.

Citations 8,9, and 10 all refer to the anti-Covid-19 vaccination. Please, specify it in the test.

Citations 9 and 10 refer to anti-COVID-19 vaccination in pregnant women and faith-based dialogue to tackle vaccine hesitancy and build trust and do not refer to the factors described in the text. 

The citation about "negative information about vaccines spread on social media" is missing (lines 56-57).

The sentence “We might reduce vaccine hesitancy by addressing these issues, but follow-up evaluations still need to be produced” in lines 58-59 is not clear. 

Citations are missing (page 2, lines 62-68)

In citation 14, the journal, volume, and page number are missing. Please, correct it.

The authors discuss the strengths of the proposed new approach, compared to individual RTCs. 

The paper does not mention systematic reviews and meta-analyses which represent the highest level of quality evidence on a research topic. 

Systematic review and Meta-analyses on the topic of vaccine acceptance and hesitancy and regarding the efficacy of interventions addressing vaccine hesitancy should be added among the references and the strengths and weaknesses with respect to the new approach proposed by the authors should be highlighted.

The authors of the manuscript refer to a mega study of Milkman KL (https:DOI: 10.1038/s41586-021-04128-4) on physical exercise but do not mention another important mega study by the same author conducted in order to encourage flu vaccination in pharmacies (https://doi: 10.1073/pnas.2115126119 ) although this study deals with a topic more relevant to the subject of the manuscript.

Vaccination hesitancy is influenced, among other things, by the clinical characteristics of the subjects; therefore, in my opinion, the clinical factors represent another important aspect of being considered and addressed in addition to social factors. Children, elderly, pregnant women, immunocompromised subjects, and subjects with chronic pathologies, are just some of the categories to which interventions should be specifically addressed.

One final comment, authors should specify the meaning of acronyms the first time they are mentioned in the text.

Reviewer 4 Report

Dear Authors

It was with great pleasure that I reviewed your manuscript.

The problem of vaccination hesitation in times of pandemic and beyond can become very serious.

I agree with you that more studies should be done on this topic. 

What you propose seems to me to be adequate.

I can tell you that I have no sequelae from polio because I was vaccinated. Fortunately, my parents decided to vaccinate me.

However, as a recommendation, I would like the conclusions to be more in-depth.

My Best Regards

Author Response

We feel great thanks for your professional review work on our article. We are so grateful for your warm comments.

We discuss the limitation of megastudies in the revision. We address the impact of social factors on vaccine hesitancy. On this basis, we discussed future directions for studying vaccine hesitancy with megastudies: introducing interventions targeting social factors, designing a quality control system, and considering justice issues in research.

We polished our conclusions to express the following opinion. Vaccine hesitancy is a typical example that reminds us that health is not just a medical problem but is closely related to society. Although the problem is complex and difficult to solve, we can still improve the situation significantly.

Thank you very much for your comment. I hope the revised manuscript could be acceptable to you.

Round 2

Reviewer 1 Report

I thank the authors for the changes made, now the paper is significantly improved. I have no other comments to offer.

Author Response

Dear Reviewer:

We are grateful for all your professional and insightful advice. You made this commentary improve significantly. 
Best wishes to you, and Happy New Year!

Sincerely,

Wai-Kit Ming, MD, PhD, MPH, MMSc

Department of Infectious Diseases and Public Health,

Jockey Club College of Veterinary Medicine and Life Sciences,

City University of Hong Kong

Address: Yuen Building, 31 To Yuen Street, Hong Kong, China

Phone: 852-34426956

Email: wkming2@cityu.edu.hk

Yibo Wu, PhD candidate, MPharm

School of Public Health, Peking University, Beijing, China

Address: 38 Xueyuan Road, Haidian District, Beijing, China

Phone: 86 18810169630

Email: bjmuwuyibo@outlook.com

Reviewer 3 Report

The authors have addressed all the comments and the manuscript has been significantly improved after the revision.

I would like to add just a small observation. The sentence of lines 228-230 could be changed as follows: “In addition, suspicion of vaccine effectiveness matters in vaccine hesitation, especially facing a new vaccine, the vaccine does not work as expected or few studies have been conducted to demonstrate its efficacy and safety in specific populations at risk" adding the reference by Zizza A et al, 2021 (Zizza, A,; Banchelli, F.; Guido, M.; Marotta, C.; Di Gennaro, F.; Mazzucco, W.; Pistotti, V.; D'Amico, R. Efficacy and safety of human papillomavirus vaccination in HIV-infected patients: a systematic review and meta-analysis. Sci Rep2021, 11(1), 4954. 

Author Response

Thank you very much for your careful check and slight revisions. Indeed, new vaccines (e.g. COVID-19) raise more concerns and hesitations than some technically mature vaccines. Therefore, we have revised the sentence as suggested (we double-checked, and that sentence is 238-240) and added the reference.
